# Ambra1 spatially regulates Src activity and Src/FAK-mediated cancer cell invasion via trafficking networks

Christina Schoenherr[1], Adam Byron[1], Emma Sandilands[1], Ketevan Paliashvili[2], George S Baillie[3,4], Amaya Garcia-Munoz[5], Cristina Valacca[6,7], Francesco Cecconi[6,7,8], Bryan Serrels[1], Margaret C Frame[1]*

[1]Cancer Research United Kingdom Edinburgh Centre, Institute of Genetics and Molecular Medicine, University of Edinburgh, Edinburgh, United Kingdom; [2]Centre for Nephrology, Division of Medicine, Royal Free Hospital Campus, London, United Kingdom; [3]Institute of Cardiovascular and Medical Science, University of Glasgow, Glasgow, United Kingdom; [4]College of Medical, Veterinary and Life Sciences, University of Glasgow, Glasgow, United Kingdom; [5]System Biology Ireland, University College Dublin, Dublin, Ireland; [6]Department of Biology, University of Tor Vergata, Via della Ricerca Scientifica, Rome, Italy; [7]Cell Stress and Survival Group, Danish Cancer Society Research Center, Copenhagen, Denmark; [8]Department of Pediatric Hematology and Oncology, IRCSS Bambino Gesu Children's Hospital, Rome, Italy

*For correspondence: m.frame@ ed.ac.uk

Competing interests: The authors declare that no competing interests exist.

**Abstract** Here, using mouse squamous cell carcinoma cells, we report a completely new function for the autophagy protein Ambra1 as the first described 'spatial rheostat' controlling the Src/FAK pathway. Ambra1 regulates the targeting of active phospho-Src away from focal adhesions into autophagic structures that cancer cells use to survive adhesion stress. Ambra1 binds to both FAK and Src in cancer cells. When FAK is present, Ambra1 is recruited to focal adhesions, promoting FAK-regulated cancer cell direction-sensing and invasion. However, when Ambra1 cannot bind to FAK, abnormally high levels of phospho-Src and phospho-FAK accumulate at focal adhesions, positively regulating adhesion and invasive migration. Spatial control of active Src requires the trafficking proteins Dynactin one and IFITM3, which we identified as Ambra1 binding partners by interaction proteomics. We conclude that Ambra1 is a core component of an intracellular trafficking network linked to tight spatial control of active Src and FAK levels, and so crucially regulates their cancer-associated biological outputs.

## Introduction

Ambra1 (activating molecule in Beclin1-regulated autophagy) is a WD40 domain-containing protein that is involved in the development of the central nervous system, adult neurogenesis and vertebrate embryogenesis (*Benato et al., 2013*; *Fimia et al., 2007*; *Yazdankhah et al., 2014*). It is an autophagy regulator, binding to Beclin1 and playing a role in the initiation of autophagy, which is required for neurogenesis (*Fimia et al., 2007*). When autophagy is not initiated, the Ambra1-Beclin1-Vps34 complex is bound to the dynein motor complex. Upon induction of autophagy, Ambra1 gets phosphorylated by the kinase ULK1, resulting in its release from the dynein complex (*Di Bartolomeo et al., 2010*; *Strappazzon et al., 2011*). Ambra1 function is negatively regulated by mTOR phosphorylation, which suppresses its binding to the E3-ligase TRAF6 and ULK1 ubiquitylation, thus

**eLife digest** In animal bodies, a mesh of proteins – known the extracellular matrix – holds cells together to give the body shape and make it more stable. Cells bind to the matrix using structures called focal adhesions. However, cells do not always stay in one place: in young animals, certain cells need to move around the body to reach their final destination. Adult animals also have some cells that are able to move, for example, to close wounds. The cells move when the focal adhesions that hold these cells in place are taken apart and then rebuilt. These processes are very dynamic and happen all the time when cells move. They are normally tightly controlled to ensure that cells only migrate under appropriate conditions. However, focal adhesions are less well regulated in cancer cells, allowing the cells to migrate away from a tumour to form new tumours elsewhere in the body.

Focal adhesions are large structures that contain many proteins. These proteins include FAK and Src, which are particularly important and have been well studied. In order to better understand how focal adhesions are taken apart, Schoenherr et al. wanted to discover new proteins that interact with FAK in skin cancer cells from mice. The experiments show that FAK binds to a protein called Ambra1, which is known to control how other proteins move around inside cells. Ambra1 and FAK work together to regulate the movement of Src away from focal adhesions and into the cell. Furthermore, Ambra1 belongs to a larger network of proteins within the cancer cells that regulates the location of Src. By changing the levels of Src and FAK at focal adhesions, Ambra1 and its other binding partners can control whether the cancer cells are attached to the extracellular matrix or are free to migrate.

Overall this work shows that the location and activity of Src within cells needs to be carefully controlled to stop the cells from moving at the wrong time. Further experiments will aim to understand which other proteins are involved in this network and how they contribute to the growth of cancer cells and the spread of tumours around the body.

controlling ULK1 stability and function (*Nazio et al., 2013*). During apoptosis, Ambra1 expression is regulated by caspase-mediated cleavage as well as degradation by calpains (*Pagliarini et al., 2012*), and a recent study reported the ubiquitylation and subsequent degradation of Ambra1 by RNF2 (*Xia et al., 2014*). Further, Ambra1 has been reported to support the binding of c-Myc to the phosphatase PP2A, leading ultimately to c-Myc degradation and reduced cell proliferation and tumourigenesis (*Cianfanelli et al., 2015*), whilst a different study implied that Ambra1 overexpression in cholangiocarcinoma has been correlated with invasion and poor survival (*Nitta et al., 2014*).

In the present study, we found that Ambra1 is a focal adhesion kinase (FAK)- and Src-binding partner. We therefore investigated the role of Ambra1 in tumour-associated phenotypes regulated by the Src/FAK pathway in squamous cell carcinoma (SCC) cells derived from the DMBA/TPA model of carcinogenesis (driven by mutated oncogenic H-Ras) (*Quintanilla et al., 1986*). We have previously shown that FAK-dependent phenotypes include cancer cell polarisation and directional migration, depending primarily on FAK's protein scaffolding activities, including binding to actin regulators like Eps8, Arp3 and RACK1 (*Schoenherr et al., 2014*; *Serrels et al., 2010*, *2007*). In particular, genetic depletion of FAK, or detachment of FAK-expressing cells, cause active Src to be trafficked from focal adhesions at the cell periphery to intracellular puncta containing autophagy proteins. We termed this 'adhesion-stress-induced autophagy', and it is FAK and adhesion dependent rather than nutrient dependent, in contrast to the classical, starvation-induced autophagy. It provides a novel mechanism for coping with high levels of active Src, and other oncogenic kinases like Ret, which are not spatially controlled in the usual way by their binding to FAK (*Sandilands et al., 2012a*, *2012b*).

We found that Ambra1 is critically involved in Src/FAK-dependent cancer cell polarisation and chemotactic invasion. In FAK-depleted SCC cells, Ambra1 controls the targeting of active Src to intracellular autophagic puncta. This is mediated by the novel Ambra1-binding proteins Dynactin 1 (Dctn1; also known as p150Glued) and interferon-induced transmembrane protein 3 (IFITM3), which together with Ambra1, are critical to trafficking processes that control spatial Src activity and Src-dependent phenotypes. A FAK mutant that is not able to bind Ambra1 promotes increased cell

adhesion and invasion by retaining more phospho-FAK (pFAK) and phospho-Src (pSrc) at focal adhesions – showing that Ambra1 can both positively and negatively regulate the amount of active Src at cellular adhesion sites. Overall, these data imply a novel role for the autophagic protein Ambra1, and its key interacting partners Dynactin 1 and IFITM3, at the heart of an intracellular trafficking network that, in turn, acts as a 'spatial rheostat' for phospho-Src and Src/FAK-mediated cancer processes.

## Results

### Ambra1 interacts with FAK in SCCs and localises at focal adhesions

We discovered Ambra1 in a phage display screen for novel FAK binding partners. As Ambra1 regulates autophagy, and we had already published that active Src was trafficked away from adhesions via autophagosomes in the absence of FAK (*Sandilands et al., 2012a*), we set out to test the involvement of Ambra1 in the autophagic trafficking of active Src. First, we confirmed Ambra1 as a novel FAK binding partner in SCC cells by reciprocal co-immunoprecipitations (co-IPs; *Figure 1A,B*).

The IPs were validated using FAK-deficient (-/-) SCC cells and Ambra1 -/- mouse embryonic fibroblasts (MEFs) (*Figure 1C*; *Figure 1—figure supplement 1A,B*). In order to confirm the co-localisation between FAK and Ambra1, we performed immunofluorescence (IF). In both SCC FAK-WT and FAK -/- cells, Ambra1 displayed diffuse cytoplasmic localisation with some at focal adhesions (*Figure 1D*). Therefore, we isolated focal adhesions from FAK-WT and FAK -/- cells by hydrodynamic force and found that Ambra1 co-localised with FAK and Paxillin at isolated focal adhesions (*Figure 1E*; *Figure 1—figure supplement 1C–E*). Additionally, Ambra1 localised to hydrodynamic force-isolated nascent focal adhesions, marked by the presence of RACK1 (*de Hoog et al., 2004*; *Serrels et al., 2010*, *2007*), in both FAK-WT and FAK -/- cells (*Figure 1F*). However, the non-autophagy, non-adhesion cellular protein CoxIV was not found at isolated focal adhesions, implying that the IF staining was not of general cellular debris (*Figure 1—figure supplement 1D,E*). In addition, we found that Ambra1 was present at focal adhesions at the cell-surface interface by Total Internal Reflection Fluorescence (TIRF) microscopy; there was partial overlap with pSrc Y416 and FAK, suggesting that, as with other autophagy proteins we have reported previously (*Sandilands et al., 2012a*; *Schoenherr et al., 2014*), there are pools of Ambra1 discretely present at sub-locations within focal adhesions (*Figure 1—figure supplement 1F,G*; *Figure 5—figure supplement 2C,D*).

### Ambra1 binds to Src and is required for trafficking of active Src to autophagosomes

We previously showed that active Src is localised to intracellular puncta containing autophagy proteins in FAK -/- SCC cells, allowing FAK-deficient cancer cells to cope with high levels of 'untethered' active Src (and other FAK-interacting tyrosine kinases, e.g. Ret) (*Sandilands et al., 2012a*, *2012b*). Other FAK- and Src-interacting proteins, like the actin regulator Eps8, are also involved in this spatial localisation of active Src (*Schoenherr et al., 2014*). Since Ambra1 is a recognised autophagy regulator, we addressed whether it was required for the trafficking of Src to autophagosomes in FAK-deficient cells. Ambra1 and Src, including active phospho-Src (pSrc; pY416), formed a biochemical complex both in FAK-expressing and -deficient SCC cells as shown by co-immunoprecipitation (*Figure 2A,B*). Using immunofluorescence, we found that Ambra1 localised to intracellular puncta that contain autophagy-regulating proteins like LC3B (*Figure 2C,D*). We noted that Ambra1 also appeared to localise to nuclei, and although we confirmed this by probing nuclear proteins isolated after cell fractionation and sucrose gradient purification of nuclei, we do not know its significance (*Figure 2—figure supplement 1A,B*). In FAK-deficient SCC cells, Ambra1 co-localised with pSrc, indicating that active Src and Ambra1 localised to the same intracellular puncta, described previously to contain autophagy proteins (*Figure 2C*). The localisation of Ambra1 in these autophagosomes was consistent with co-immunoprecipitation with LC3B, providing evidence that they were present in the same biochemical complex (*Figure 2E*). We note that there was a reduced steady state level of the lipidated form of LC3B (LC3B II) in the SCC FAK -/- cells (the significance of this is unknown; [*Sandilands et al., 2012a*]). We are not able to discern which isoform of LC3B is binding to Ambra1 or whether there is any difference between them.

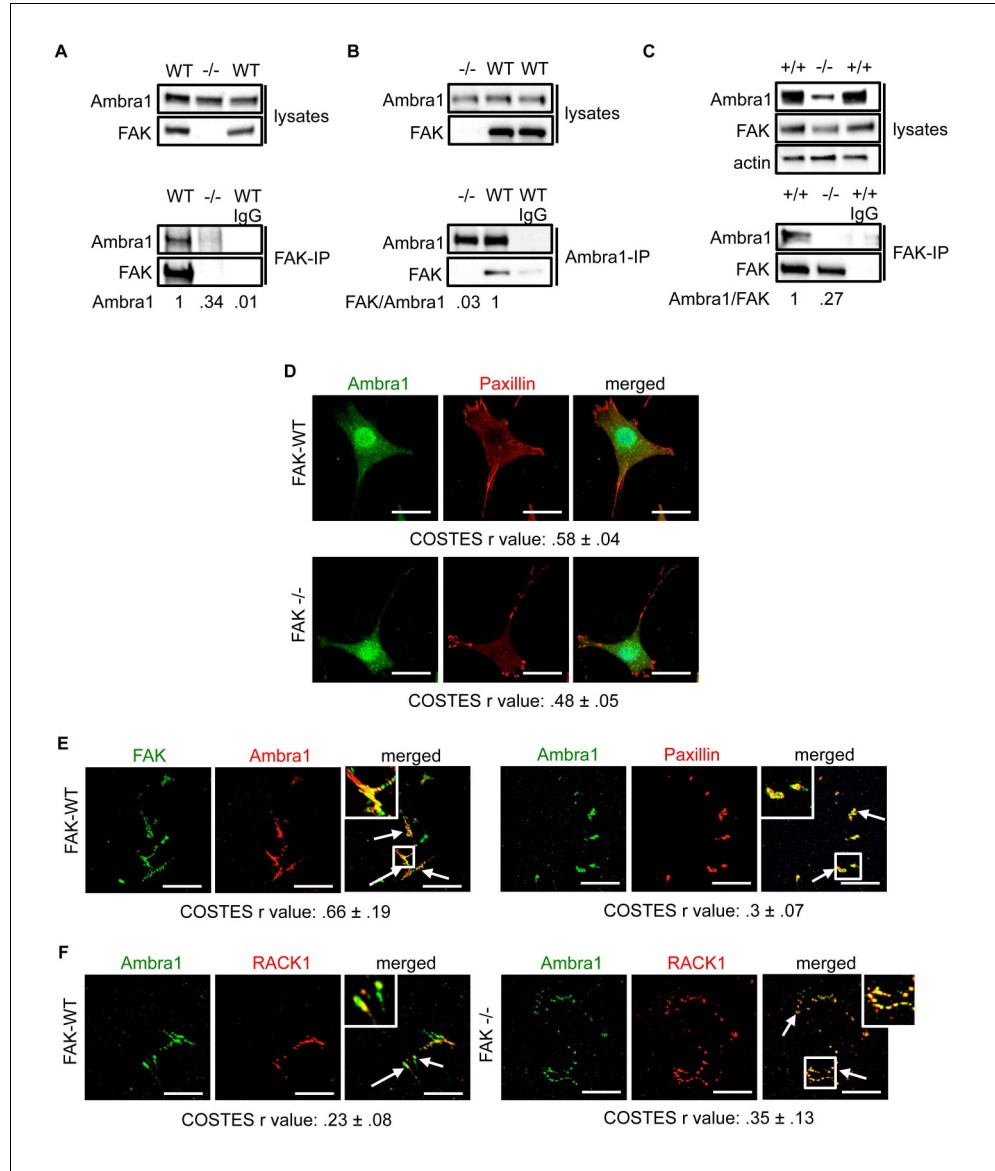

**Figure 1.** Ambra1 interacts with FAK at focal adhesions. FAK or Ambra1 were immunoprecipitated from FAK-WT and FAK -/- cell lysates using (**A**) anti-FAK 4.47 agarose or (**B**) anti-Ambra1, followed by western blot analysis with anti-FAK and anti-Ambra1. (**C**) FAK was immunoprecipitated from Ambra1 +/+ and Ambra1 -/- MEF cell lysates using anti-FAK 4.47 agarose, followed by western blot analysis with anti-FAK and anti-Ambra1. Anti-$\beta$-actin was used as a loading control. Relative ratios of Ambra1, FAK/Ambra1 and Ambra1/FAK interactions were calculated by densitometry. (**D**) FAK-WT and FAK -/- cells were seeded onto glass coverslips, fixed and stained using anti-Ambra1 and anti-Paxillin. Scale bars, 20 μm. (**E, F**) Focal adhesions were isolated from FAK-WT and FAK -/- cells using hydrodynamic force. (**E**) Focal adhesions (solid arrows) were stained with anti-FAK and anti-Ambra1 (left panels) and with anti-Ambra1 and anti-Paxillin (right panels). (**F**) Focal adhesions (solid arrows) were stained with anti-Ambra1 and anti-Rack1 in SCC FAK-WT (left panels) and SCC FAK -/- cells (right panels). Scale bars, 20 μm. Colocalisation (Costes *r* value from five cells) was analysed using the ImageJ plugin JaCoP (*Bolte and Cordelières, 2006*).

The following source data and figure supplements are available for figure 1:

**Source data 1.** COSTES r values for immunofluorescence images.

**Figure supplement 1.** Ambra1 +/+ and -/- mouse embryonic fibroblasts (MEFs).

*Figure 1 continued on next page*

*Figure 1 continued*

**Figure supplement 2.** Knockdown of Ambra1 suppresses FAK phenotypes.

To investigate whether Ambra1 controlled active Src localisation, we suppressed expression of Ambra1 with both a pool of four and two individual siRNAs (*Figure 2F*). Knockdown of endogenous Ambra1 either by the pool of four siRNAs (*Figure 2G,H*) or two individual siRNAs (*Figure 2I,J*) resulted in the redistribution of active Src from intracellular puncta back to focal adhesions in FAK -/- cells. However, we were unable to determine whether this relocation would also occur in detached cells, which do not have focal adhesions. Taken together, these data imply that Ambra1 is required for the trafficking of active Src from focal adhesions to intracellular autophagic puncta.

## Ambra1 is part of a network of trafficking proteins linked to active Src and FAK

We next performed Ambra1, pSrc Y416, and FAK (and IgG control) IPs using lysates from FAK-WT and FAK -/- cells (in triplicate) and determined specifically interacting proteins by quantitative label-free mass spectrometry. Amongst the binding partners of all three 'baits' were proteins that are involved in regulating intracellular trafficking, raising the possibility that Ambra1 is part of a Src/FAK 'trafficking network' (*Figure 3A*; *Supplementary file 1–3*).

To test the hypothesis that Ambra1-dependent targeting of pSrc from focal adhesions to intracellular autophagic puncta involved such trafficking proteins, we selected two novel Ambra1 binding partners for further study. These were Dynactin 1, whose interaction with Ambra1 appeared to be enriched in FAK-deficient cells from proteomics analyses (*Figure 3*) and IFITM3, which was positioned at the centre of the trafficking network as it bound to all three of Ambra1, pSrc and FAK (*Figure 3*). Dctn1 (Dynactin 1) is a component of the dynactin complex and is involved in dynein-mediated transport along microtubules, enhancing motor processivity (*Culver-Hanlon et al., 2006*; *King and Schroer, 2000*; *Waterman-Storer et al., 1995*), and mediating trafficking and maturation of EGFR and lysosomes (*Kedashiro et al., 2015b*; *Li et al., 2014*). We confirmed the interaction between Ambra1 and Dynactin 1 by co-IP, thereby validating the interaction identified by mass spectrometry (*Figure 3B*). IFITM3, which forms a central node linking the Ambra1, pSrc and FAK trafficking networks (*Figure 3A*), is known to block cellular entry of viruses (*Amini-Bavil-Olyaee et al., 2013*; *Brass et al., 2009*; *Feeley et al., 2011*; *Huang et al., 2011*; *Lu et al., 2011*; *Weidner et al., 2010*), to traffic with Rab7 and LAMP1, and to interact with v-ATPase on endosomes to enable clathrin localisation (*Feeley et al., 2011*; *Wee et al., 2012*). IFITM3 is highly expressed in many cancers (*Andreu et al., 2006*; *Hu et al., 2014*; *Li et al., 2011*; *Yang et al., 2013*) and has been variously implicated in tumour cell proliferation, migration and invasion (*Hu et al., 2014*; *Li et al., 2011*; *Yang et al., 2013*; *Zhao et al., 2013*). We confirmed the interaction between Ambra1 and IFITM3 (*Figure 3C*), FAK and IFITM3 (*Figure 3D*), and pSrc and IFITM3 (*Figure 3E*) by co-IPs. The interactions between IFITM3 and both Ambra1 and pSrc appeared to be enriched in FAK-expressing cells due to increased IFITM3 expression levels, caused by FAK-dependent IFITM3 transcription (not shown; (*Figure 3A*), and we believe that IFITM3 is present at focal adhesions, shown by immunoblotting of isolated focal adhesion preparations (*Figure 3—figure supplement 1A*).

To determine whether IFITM3 was required for complex formation between active Src and Ambra1 in FAK-WT and -/- cells, we knocked down IFITM3 using a pool of siRNAs and analysed the interaction of Ambra1 and pSrc by co-immunoprecipitations (*Figure 3—figure supplement 1B,C*). While knockdown of IFITM3 did not affect the Ambra1-pSrc interaction (*Figure 3—figure supplement 1B,C*), IFITM3 depletion reduced the interaction between FAK and active pSrc, indicating that IFITM3 is involved in the optimal association of Src to FAK (*Figure 3—figure supplement 1D,E*), likely through regulating their precise localisation. Thus, interaction network analysis coupled with co-immunoprecipitations suggest that IFITM3 is a central component of a trafficking network linking Ambra1, FAK and active Src (*Figure 3A*).

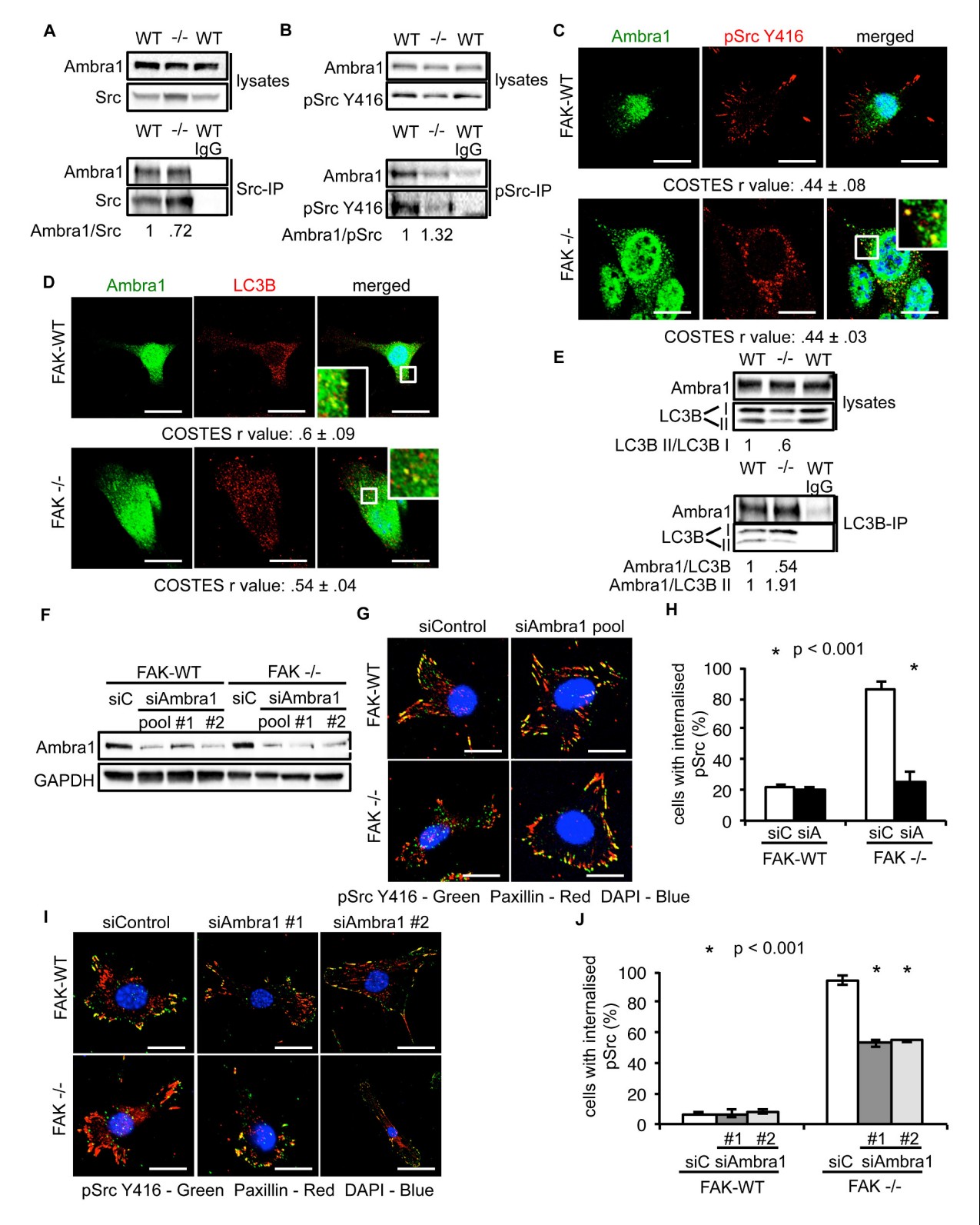

**Figure 2.** Ambra1 interacts with Src and mediates trafficking of active Src to autophagosomes. (**A, B**) Src (**A**) or pSrc Y416 (**B**) were immunoprecipitated from FAK-WT and FAK -/- cell lysates using anti-Src agarose or anti-pSrc Y416 antibody, followed by western blot analysis with anti-Ambra1, anti-pSrc Y416 and anti-Src. Relative ratios of Ambra1/Src and Ambra1/pSrc interactions were calculated by densitometry. (**C**) FAK-WT and FAK -/- cells were seeded onto glass coverslips, fixed and stained using anti-pSrc Y416, anti-Ambra1 and DAPI. Scale bars, 20 μm. (**D**) SCC FAK-WT and FAK -/- cells were

*Figure 2 continued on next page*

*Figure 2 continued*

grown on glass coverslips, fixed and stained with anti-Ambra1, anti-LC3B and DAPI. Scale bars, 20 μm. (E) LC3B was immunoprecipitated from SCC FAK-WT and FAK -/- cell lysates using anti-LC3B, followed by western blot analysis with anti-Ambra1 and anti-LC3B. Relative ratios of LC3B II/LC3B I as well as the Ambra1/LC3B and Ambra1/LC3B II interactions were calculated by densitometry. (F–J) SCC FAK-WT and FAK -/- cells were transiently transfected with either a pool (F–H) or two individual siAmbra1 siRNAs (F, I, J). The cells were grown on glass coverslips, fixed and stained with anti-pSrc Y416, anti-Paxillin and DAPI. (G, I) Representative immunofluorescence images. Scale bars, 20 μm. (H, J) Quantification of internalised active Src. n = 3. Error bars, s.d. p<0.001. Colocalisation (Costes *r* value from five cells) was analysed using the ImageJ plugin JaCoP.

The following source data and figure supplement are available for figure 2:

**Source data 1.** COSTES r values for immunofluorescence images and percentage of cells with internalised pSrc.

**Figure supplement 1.** Ambra1 interacts with Src and mediates pSrc trafficking.

## Dynactin 1 and IFITM3 are involved in the trafficking of pSrc to autophagosomes

Having verified that both Dynactin 1 and IFITM3 bind to Ambra1, we next addressed whether they were involved in the Ambra1-dependent intracellular trafficking of active pSrc. We transiently knocked down endogenous Dynactin 1 (*Figure 4A*) or IFITM3 (*Figure 4D*) by siRNA, and analysed the localisation of active pSrc in FAK-deficient SCC cells. In both cases, suppression of protein expression resulted in significant redistribution of active pSrc from intracellular autophagic puncta to focal adhesions (*Figure 4B,C*; *Figure 4—figure supplement 1A* and *Figure 4E,F*; *Figure 4—figure supplement 1B* respectively), while in FAK-expressing cells, knockdown of Dynactin 1 or IFITM3 had no visible effect on their distribution. These data imply that Ambra1 is part of a functional trafficking network that precisely regulates the spatial distribution of Src activity. Mechanistically, Ambra1 functions *via* interaction with proteins involved in intracellular trafficking, including Dynactin 1 and IFITM3; the latter lies at the centre of linked Ambra1, pSrc, and FAK interactomes.

## Ambra1 is required for Src/FAK-mediated cancer-related processes

Having established that Ambra1 and FAK interact with each other and co-localise at focal adhesions (*Figure 1*), and that they are co-determinants of the intracellular localisation of Src activity, we next addressed whether Ambra1 was required for Src/FAK-mediated cancer-related processes. We therefore knocked down Ambra1, and found that there was significant loss of polarisation towards a monolayer wound in FAK-expressing SCC cells, as judged by the direction of the Golgi apparatus (stained with GM130). This was in contrast to FAK-deficient cells, in which knockdown of Ambra1 had no further effect on the already suppressed polarisation of cells towards a wound (*Figure 1—figure supplement 2A,B*). The role of Ambra1 in polarisation was confirmed using Ambra1 +/+ and -/- MEFs (*Figure 1—figure supplement 2C*). In addition, we found that Ambra1 was required for chemotactic invasion into growth factor-reduced Matrigel in FAK-expressing cells (*Figure 1—figure supplement 2D,E*), whilst there was no effect in FAK-deficient cells, which, as we described previously, do not invade into Matrigel (*Serrels et al., 2012*). These results describe a previously unknown role for Ambra1 in cancer cell polarisation and invasion that is Src/FAK dependent.

### A FAK Ambra1-binding mutant increased adhesion and active Src at adhesions

As Ambra1 binds FAK and is found at isolated focal adhesions, we next addressed whether FAK and Ambra1 interact directly. The Ambra1 binding site in FAK was mapped by peptide array binding analysis as described previously (*Schoenherr et al., 2014*; *Serrels et al., 2007*), which resulted in identification of two amino acids in FAK that were required for optimal direct binding of Ambra1, i.e. amino acids P875 and P881. These prolines (P) were mutated to alanines (A), and the resulting FAK P875A/P881A (AA) mutant caused reduced binding of Ambra1 to FAK within cells (*Figure 5A, B*). The interaction between FAK and p130Cas, which also binds FAK at a similar proline-rich region but not the same combination of amino acids (P715, P718, P878 and P881; ref. [*Harte et al., 1996*]), was more modestly affected by the P875A/P881A mutation (quantified in *Figure 5C*).

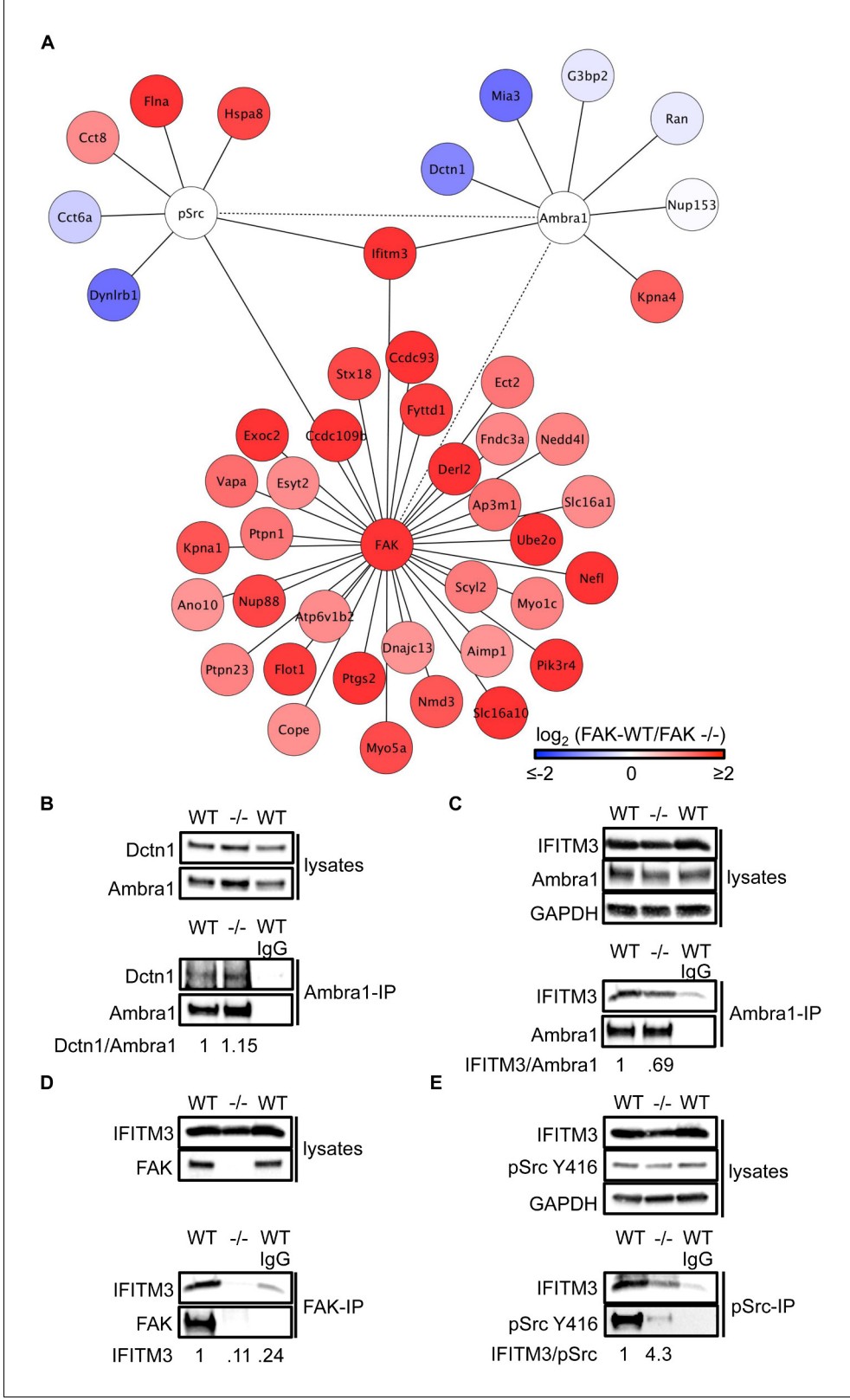

**Figure 3.** IFITM3 is in the centre of an Ambra1, FAK and pSrc trafficking network. (**A**) Network analysis of Ambra1-, FAK- and pSrc Y416-interacting proteins that are involved in trafficking processes. Solid lines indicate protein-protein interactions identified in the mass spectrometry datasets used for the interaction map. Dotted lines indicate Ambra1–FAK/pSrc interactions, which have been previously identified and verified by immunoprecipitation. (**B**) Ambra1 was immunoprecipitated from FAK-WT and FAK -/- cell lysates using anti-Ambra1 antibody, followed by western blot analysis with anti-

*Figure 3 continued on next page*

*Figure 3 continued*

Dctn1 and anti-Ambra1. (C–E) Ambra1 (C), FAK (D) or pSrc Y416 (E) were immunoprecipitated from FAK-WT and -/- cell lysates, followed by western blot analysis with anti-IFITM3, anti-Ambra1, anti-FAK and anti-pSrc Y416. Anti-GAPDH served as a loading control. Relative ratios of Dctn1/Ambra1, IFITM3/Ambra1, IFITM3 and IFITM3/pSrc interactions were calculated by densitometry.

The following figure supplement is available for figure 3:

**Figure supplement 1.** IFITM3 influences the Src-FAK complex.

When culturing the cells expressing FAK that was impaired in Ambra1 binding, we noticed that these FAK P875A/P881A cells seemed to display greater adherence than the FAK-WT cells. Therefore, we performed adhesion assays on fibronectin-coated dishes (*Figure 5D*) or on plastic (*Figure 5—figure supplement 1A*). We found that after 20 and 60 min, FAK P875A/P881A and FAK -/- cells attached to a higher degree than FAK-WT cells. Furthermore, in already adhered FAK P875A/P881A cells (*Figure 5E*, middle panels), we found that there was more intense pSrc staining at focal adhesions when compared to FAK-WT cells (*Figure 5E*, top panels) or FAK -/- cells, in which pSrc was present at intracellular autophagic puncta (*Figure 5E*, lower panels). Quantification of the relative intensity of pSrc at focal adhesions (*Figure 5F*) and of pSrc in intracellular puncta (*Figure 5—figure supplement 1B*) is shown. While there was more intense staining of active pSrc at focal adhesions in FAK P857A/P881A cells, there were no significant changes in the number or size of focal adhesions (*Figure 5—figure supplement 1C,D*). The increased staining of pSrc (and pFAK Y397; *Figure 5—figure supplement 1E*) in FAK P875A/P881A cells was confirmed by immunoblotting isolated focal adhesion preparations (*Figure 5G*, right panels, red dots; quantified as phospho/total Src (and phospho/total FAK) in *Figure 5H*), demonstrating that both active pSrc and pFAK were elevated relative to total Src and FAK, respectively. This was true also of pPaxillin Y118 (*Figure 5G*, right panels), reflecting specific retention of phospho- and activated- focal adhesion components at focal adhesions when FAK cannot bind to Ambra1. This implies that the FAK–Ambra1 complex is crucial to orchestrate the selective removal of active focal adhesion components from focal adhesions or to promote their turnover at these adhesion sites. The elevated levels of active components retained at focal adhesions when the binding of FAK to Ambra1 is impaired may contribute to the enhanced adhesion at early times after plating on extracellular matrix (*Figure 5D*).

## Ambra1 binding-deficient FAK causes increased invasion and 3D proliferation

To determine whether there were other biological consequences of the aberrant accumulation, or retention, of active pSrc and pFAK at focal adhesions when FAK binding to Ambra1 was impaired, we examined chemotactic invasion into growth factor-reduced Matrigel and proliferation in three-dimensional (3D) culture. We found that elevated active pSrc and pFAK at focal adhesions in FAK P875A/P881A cells was associated with enhanced invasive migration (*Figure 6A,B*) and increased proliferation in 3D compared to FAK-WT cells (*Figure 6C,D*), without affecting the Src-FAK interaction (*Figure 5—figure supplement 2E*). As we found the Ambra1 binding partners Dynactin 1 and IFITM3 to regulate the trafficking of active Src in FAK-deficient SCC cells (*Figure 4*), we asked whether they also influenced mis-regulation of active pSrc and pFAK levels at focal adhesions when FAK can no longer bind to Ambra1. We therefore transiently knocked down endogenous Dynactin 1 or IFITM3 expression using siRNA, and we found that the 'over-invasion' of FAK P875A/P881A-expressing cells was restored to similar, or lower, invasion levels than FAK-WT-expressing cells (*Figure 6E–G*). Additionally, elevated active pSrc levels in the FAK P875A/P881A SCC cells were restored to normal (wildtype) levels by knocking down Dynactin 1 (not shown). These results demonstrate that Ambra1 binding to FAK is required for the maintenance of normal steady-state levels of active pSrc and pFAK at focal adhesions. When this binding is perturbed so that pSrc and pFAK are elevated, the resultant enhanced invasion requires the Ambra1-interacting trafficking proteins Dynactin 1 and IFITM3, which are therefore crucial mediators of active Src trafficking.

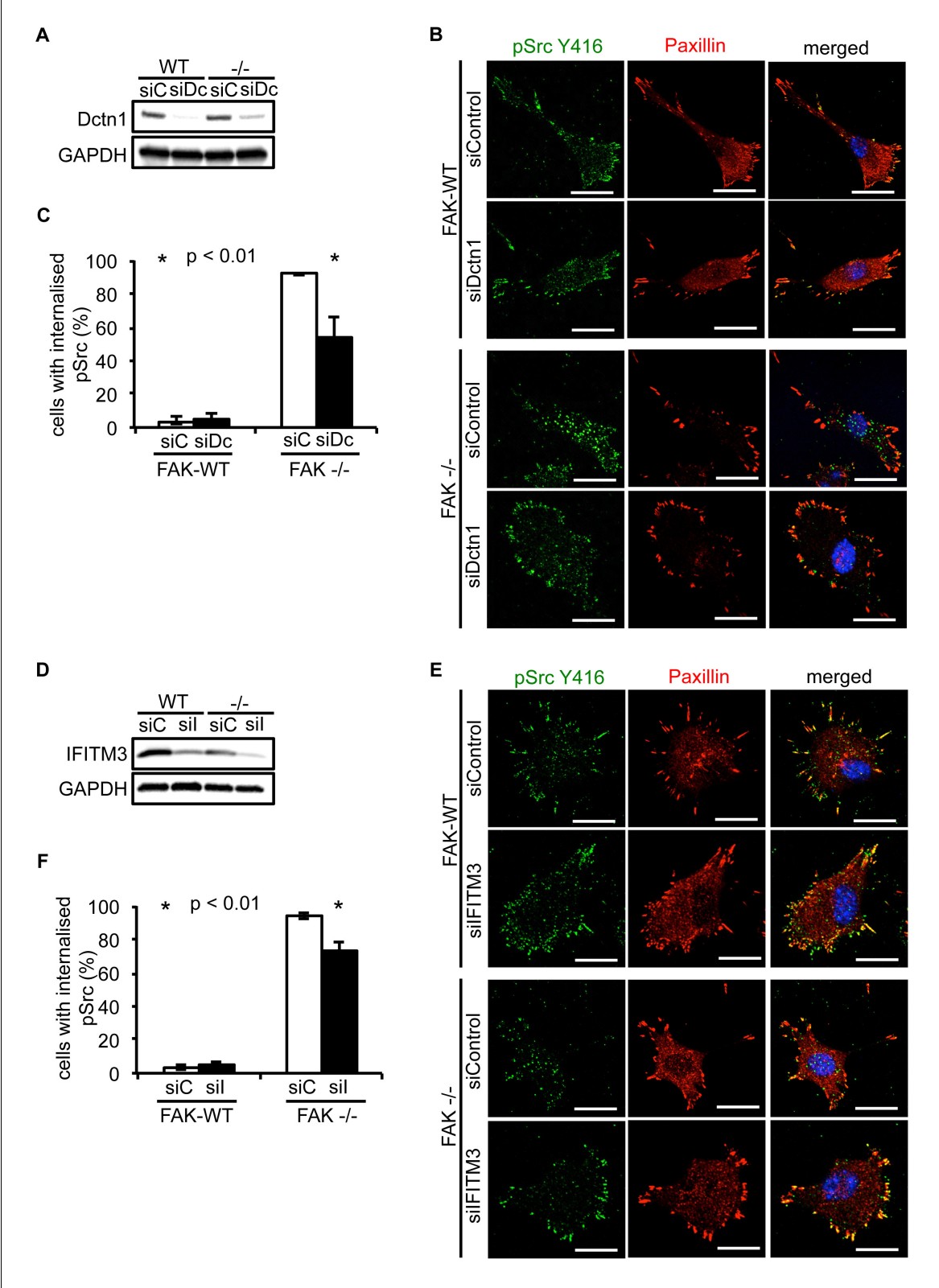

**Figure 4.** Knockdown of Dynactin 1 and IFITM3 suppresses trafficking of active Src to autophagic puncta. (**A**) FAK-WT and FAK -/- cells were transiently transfected with a pool of Dynactin 1 (Dctn1) siRNAs and lysed 48 hr post transfection. Dynactin 1 expression was determined by western blotting using anti-Dctn1. Anti-GAPDH was used as a loading control. (**B**) SCC FAK-WT and FAK -/- cells transiently transfected with Dctn1 siRNA were grown on glass coverslips, fixed and stained with anti-pSrc Y416, anti-Paxillin and DAPI. Scale bars, 20 μm. (**C**) Quantification of internalised active Src. *n* = 3. Error bars,

*Figure 4 continued on next page*

*Figure 4 continued*

s.d. p<0.01. (D) FAK-WT and FAK -/- cells were transiently transfected with a pool of IFITM3 siRNAs and lysed 48 hr post transfection. IFITM3 expression was determined by western blotting using anti-IFITM3. Anti-GAPDH was used as a loading control. (E) SCC FAK-WT and FAK -/- cells transiently transfected with IFITM3 siRNA were grown on glass coverslips, fixed and stained with anti-pSrc Y416, anti-Paxillin and DAPI. Scale bars, 20 μm. (F) Quantification of internalised active Src. *n* = 3. Error bars, s.d. p<0.01.
The following source data and figure supplement are available for figure 4:

**Source data 1.** Percentage of cells with internalised pSrc.
**Figure supplement 1.** Knockdown of Dynactin 1 and IFITM3 suppress pSrc trafficking to autophagic puncta.

## Discussion

### Ambra1 is involved in Src/FAK-mediated cancer phenotypes

Our results define Ambra1 as a crucial regulator of Src/FAK-mediated cancer phenotypes, acting as a 'spatial rheostat' to maintain steady-state levels of active pSrc and pFAK at adhesions (see model in *Figure 7*). When FAK is present, Ambra1 forms a complex with FAK and Src and plays a key a role in Src/FAK-regulated cancer cell adhesion, polarisation and invasion (*Figure 7A*). In FAK-deficient cells, Ambra1 controls the targeting of active Src away from focal adhesions into autophagic structures that the cells use to cope with toxic levels of active Src when it is not 'tethered' at adhesion sites by FAK (*Figure 7B*; *Sandilands et al., 2012a*). When Ambra1 is unable to bind to FAK (as is the case in FAK P875A/P881A-expressing cells), this causes over-retention (lack of removal or turnover) of active Src (and FAK) from focal adhesions, promoting cell adhesion and enhanced invasive migration (*Figure 7C*). Ambra1 is therefore a novel spatial regulator of the active Src/FAK complex at sites of cell-matrix adhesion, controlling downstream biological effects in cancer cells. It does this by scaffolding trafficking proteins, for example, Dynactin 1 and IFITM3, which themselves lie at the heart of a network of scaffolding proteins that are needed for optimal trafficking of active Src (and FAK), maintaining appropriate and tolerated levels.

### Control of Src/FAK spatial activities is a novel Ambra1 function

Ambra1 is crucial for the development of the central nervous system, neurogenesis and embryogenesis (*Benato et al., 2013*; *Fimia et al., 2007*; *Yazdankhah et al., 2014*), and is an autophagy regulator, via its interaction with Beclin1 (*Fimia et al., 2007*). The role of Ambra1 in cancer is less clear. One study suggested that Ambra1 may have a tumour suppressor role, as it is involved in the degradation of c-Myc induced by dephosphorylation by PP2A, resulting in reduced proliferation and tumourigenesis (*Cianfanelli et al., 2015*). However, another study implied that Ambra1 may have tumour-promoting functions, since Ambra1 overexpression correlated with invasion and poor survival in cholangiocarcinoma (*Nitta et al., 2014*). The new findings we present here show that Ambra1 contributes to SCC cell behaviour by scaffolding both Src and FAK, and intracellular trafficking proteins, controlling steady-state levels and spatial distribution of the activated forms of Src and FAK at focal adhesions. In this context, therefore, Ambra1 is a scaffold protein that promotes cancer cell phenotypes driven by the Src/FAK pathway, including cancer cell polarisation and chemotactic invasion. This central trafficking scaffold function of Ambra1, and the discovery that it is a crucial 'spatial rheostat' for Src/FAK signalling, adds to the number of important physiological functions for Ambra1, and defines a new level of regulation of the Src/FAK signalling axis. It is interesting to note that while knockdown of Ambra1 caused reduced cancer cell polarisation towards the denuded area of a wounded monolayer, and suppressed chemotactic invasive migration in FAK-expressing cancer cells, there was no effect in FAK-deficient cells, which display intrinsically low polarisation and invasion capacity. We conclude that Ambra1 is a FAK scaffold required for FAK-dependent cancer cell phenotypes, most likely by controlling the localisation of active FAK, and its upstream regulator Src, at focal adhesions.

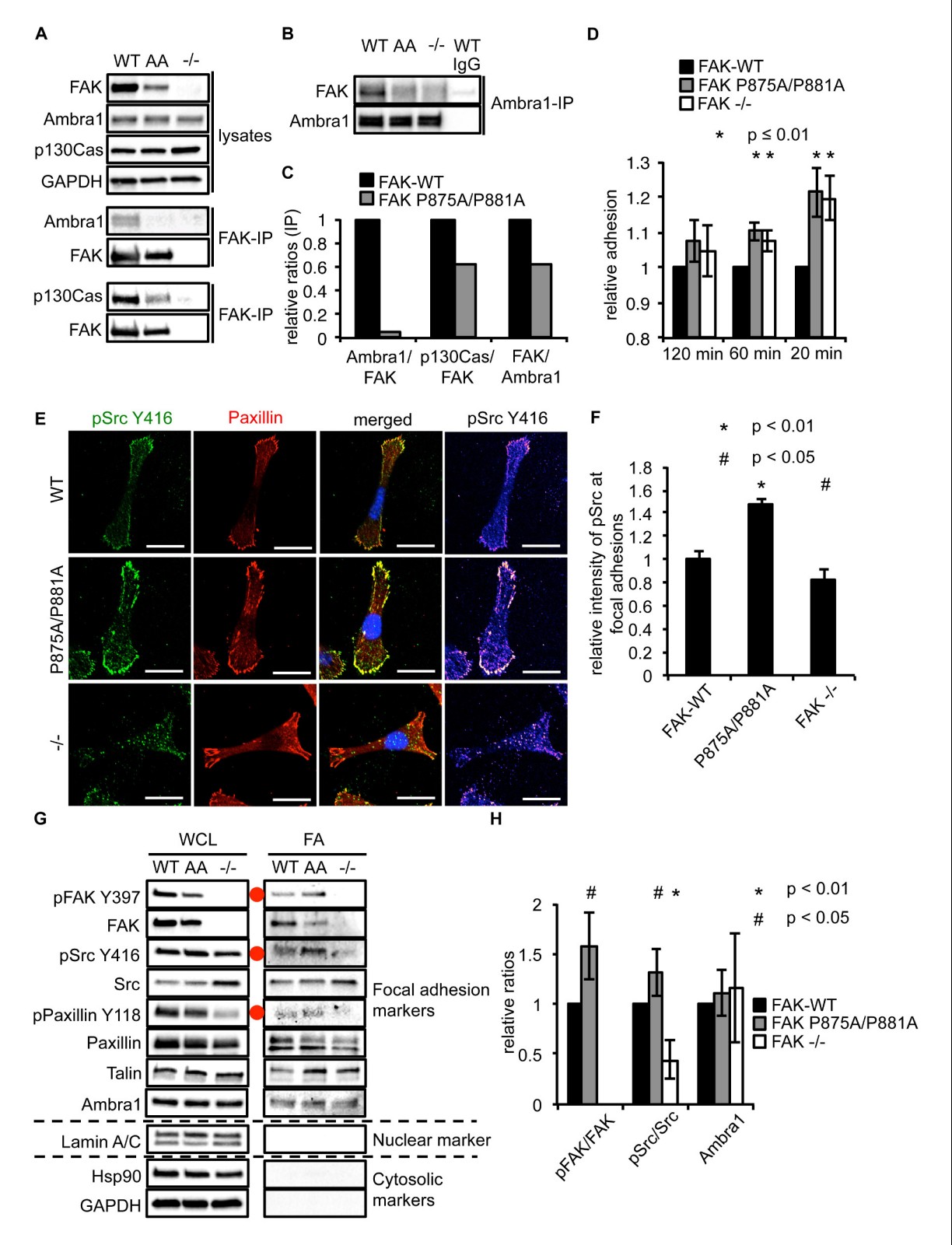

**Figure 5.** Ambra1 binding impaired FAK increases cell adhesion and pSrc at focal adhesions. (A, B) FAK (A) or Ambra1 (B) were immunoprecipitated from FAK-WT, FAK P875A/P881A (AA) and FAK -/- cell lysates using anti-FAK 4.47 agarose or anti-Ambra1, followed by western blot analysis with anti-FAK, anti-Ambra1 and anti-p130Cas. Anti-GAPDH was used as a loading control. (C) Relative ratios of Ambra1/FAK and p130Cas/FAK interactions were calculated by densitometry. (D) Adhesion assay: SCC FAK-WT, FAK P875A/P881A and FAK -/- cells were plated in serum-free conditions on fibronectin-
*Figure 5 continued on next page*

*Figure 5 continued*

coated plates. Samples were normalised to the 6 hr time point and relative adhesion was calculated by setting the FAK-WT values to 1. *n* = 3. Error bars, s.d. p≤0.01. (**E, F**) SCC FAK-WT, FAK P875A/P881A and FAK -/- cells were grown on glass coverslips for 24 hr, fixed and stained with anti-pSrc Y416, anti-Paxillin and DAPI. Scale bars, 20 μm. The relative intensity of pSrc staining at focal adhesions from five cells (at least 10 focal adhesions/cell) was measured using ImageJ. (**E**) Representative immunofluorescence images are shown. (**F**) Quantification of the relative pSrc intensity at focal adhesions. *n* = 5. Error bars, s.e.m. p<0.01 (*) and p<0.05 (#). (**G**) Focal adhesions from SCC FAK-WT, FAK P875A/P881A (AA) and FAK -/- cells were crosslinked and isolated for western blot analysis with the indicated antibodies. Paxillin and Talin served as markers for focal adhesions and Lamin A/C was used as a nuclear marker. Hsp90 and GAPDH represented cytosolic markers. WCL, whole cell lysate; FA, focal adhesions. The purity of the isolated focal adhesions was determined by the absence of nuclear proteins like Lamin A/C and cytosolic markers like Hsp90 and GAPDH. There was less active Src at focal adhesions in the FAK-deficient SCC cells due to Src's internalisation to autophagic structures. Additionally, increased pPaxillin Y118 and Talin levels could be detected in the FAK Ambra1-binding mutant compared to SCC FAK-WT and FAK -/- cells. No changes in Ambra1 levels at focal adhesions could be detected. (**H**) The relative ratios of pFAK/FAK, pSrc/Src and Ambra1 at focal adhesions were calculated using densitometry. *n* = 3. Error bars, s.d. p<0.01 (*) and p<0.05 (#).

The following source data and figure supplements are available for figure 5:

**Source data 1.** Relative mean values of adhesion, pSrc intensity at focal adhesions and relative ratios at focal adhesions.
**Figure supplement 1.** FAK Ambra1-binding mutant promotes adhesion and invasion.
**Figure supplement 2.** FAK Ambra1-binding mutant promotes adhesion and invasion.

## Ambra1 is at the centre of a trafficking network that spatially regulates Src

Ambra1 is reported to be an autophagy regulator (*Fimia et al., 2007*), and we found that it is crucially involved in the redistribution of active Src away from focal adhesions into intracellular autophagic puncta in FAK-deficient cancer cells (described previously in *Sandilands et al., 2012a*). Ambra1 is localised to intracellular puncta containing the autophagosomal marker LC3B, and it interacts with LC3B. These data implicate Ambra1 in the selective autophagic trafficking of active Src when cancer cells are under severe 'adhesion stress', such as when FAK is absent.

We found that Ambra1-interacting proteins (identified by mass spectrometry) included the dynactin complex component Dynactin 1 and IFITM3, in addition to other proteins considered to be intracellular trafficking proteins. As Ambra1 has been reported to interact with the dynein complex, we thought it likely that Ambra1 may regulate the trafficking of active Src to intracellular autophagic puncta *via* dynein-dependent processes (*Di Bartolomeo et al., 2010*), and we found that knockdown of endogenous Dynactin 1 by siRNA inhibited the trafficking of active Src from focal adhesions to autophagosomes. Dynactin 1 is thought to be involved in retrograde trafficking and maturation of trafficking vesicles (*Jovasevic et al., 2015*; *Kedashiro et al., 2015a*; *Li et al., 2014*; *Ohbayashi et al., 2012*; *Verissimo et al., 2015*), implying that these may be involved in the redistribution of active Src from focal adhesions to autophagosomal puncta in FAK-deficient cells. We noted that Dynactin 1 knockdown resulted in enlarged late endosomes (not shown), while the number of late endosomes was not altered (not shown), implying that impaired trafficking/maturation of vesicles may be responsible for the block to autophagic targeting of active Src upon Ambra1 and Dynactin 1 knockdown. Further, IFITM3 has been reported to localise to late endosomes/lysosomes and overexpression expands Rab7- and LAMP1-positive structures (*Feeley et al., 2011*), and loss of IFITM3 blocks clathrin-mediated phagocytosis by removing clathrin from membranes (*Wee et al., 2012*). Knockdown of IFITM3 also significantly impaired the trafficking of active Src to autophagic puncta. In proteomic network analyses, we found that IFITM3 interacts both with FAK and active Src, as well as Ambra1, thereby linking the Ambra1-, FAK- and pSrc-interacting trafficking protein networks together. Furthermore, IFITM3 appears to be important for the interaction of active pSrc with FAK as well as to a lesser extent of Ambra1 with FAK (not shown), but not Ambra1 with pSrc. These findings suggest that IFITM3 is involved in regulating FAK's interaction with Ambra1 and pSrc. Overall, these data place IFITM3 at the centre of a larger trafficking protein network linking Ambra1, FAK and active Src sub-networks. The full range of the Src/FAK/Ambra1-interacting trafficking proteins from the interactome network (*Figure 3*) that play a role in pSrc trafficking are unknown at present.

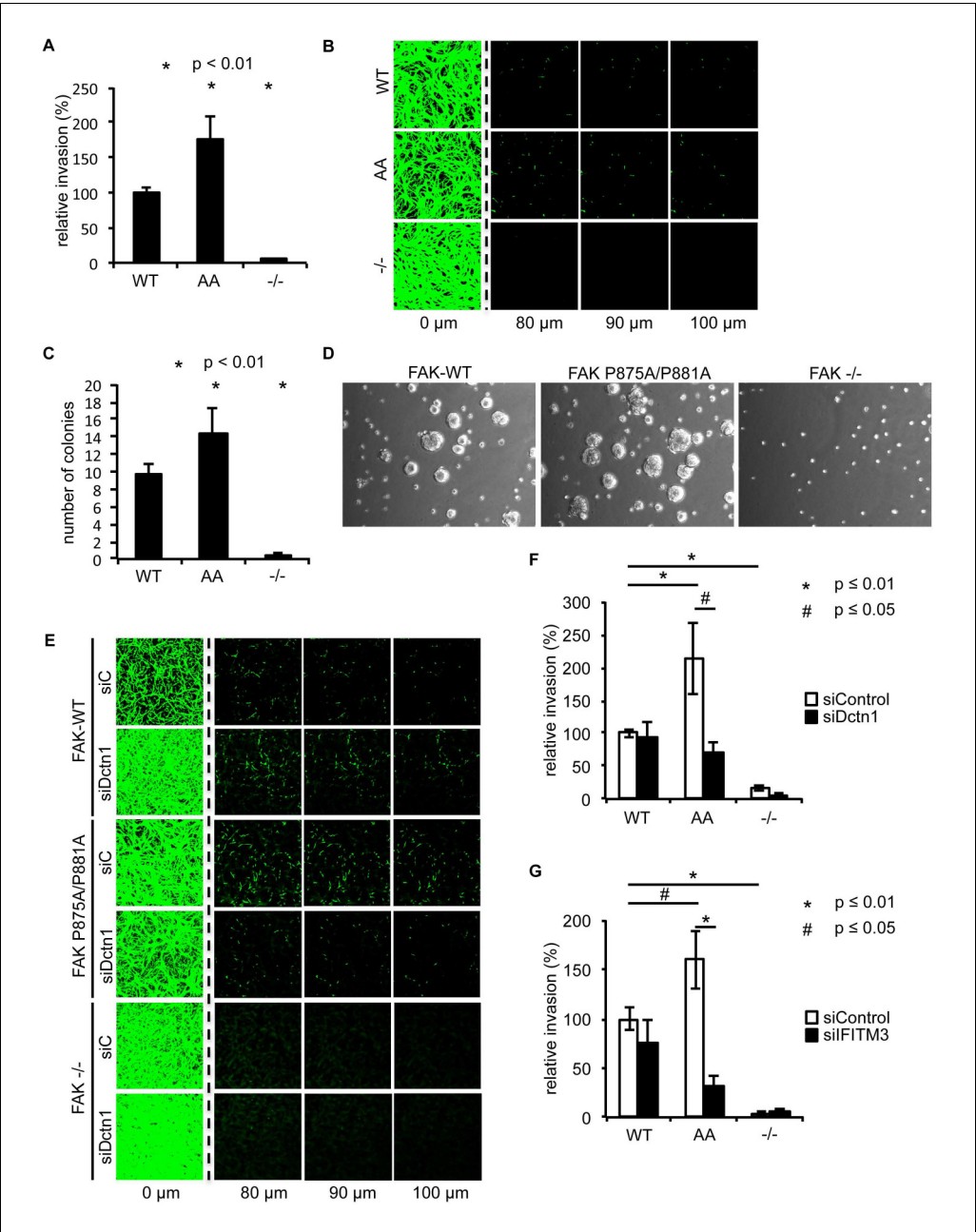

**Figure 6.** Ambra1 binding-impaired mutant FAK increases invasion and 3D proliferation, which is rescued by reducing Dynactin 1 or IFITM3 expression levels. (**A**) Invasion assay: SCC FAK-WT, FAK P875A/P881A (AA) and FAK -/- cells were seeded on growth factor-reduced Matrigel in serum-free conditions. After 72 hr, invasion towards a serum gradient was visualised by staining the cells with calcein. $n = 8$. Error bars, s.e.m. $p\leq0.01$. (**B**) Representative images of the invasion assay. (**C, D**) 3D proliferation assay: SCC FAK-WT, FAK P875A/P881A and FAK -/- cells were resuspended in methylcellulose solution in growth medium on a layer of agarose. After nine days, images were taken from 6–10 random fields and colonies were counted. (**C**) Quantification of the 3D proliferation assay. $n = 3$. Error bars, s.d. $p<0.01$. (**D**) Representative images of the 3D proliferation assay. (**E–G**) Invasion assay: SCC FAK-WT, FAK P875A/P881A and FAK -/- cells transiently transfected with Dctn1 siRNA were seeded on growth factor-reduced Matrigel in serum-free conditions. After 72 hr, invasion towards a serum gradient was visualised by staining the cells with calcein. (**E**) Representative images of the invasion assay. (**F**) Quantification of the invasion assay. $n = 6$. Error bars, s.e.m. $p\leq0.01$. (**G**) Quantification of the invasion assay with cells transiently transfected with IFITM3 siRNA. $n = 6$. Error bars, s.e.m. $p\leq0.01$ (*) and $p<0.05$ (#).

*Figure 6 continued on next page*

*Figure 6 continued*

The following source data is available for figure 6:

**Source data 1.** Mean values of invasion and number of colonies.

## The FAK-Ambra1 complex controls localisation of active adhesion components

Expression of a mutant FAK protein (in otherwise FAK-deficient cancer cells) that was hugely impaired in its ability to bind Ambra1 (FAK P875A/P881A), demonstrated the importance of the complex between FAK and Ambra1 in promoting the removal of phosphorylated components from focal adhesions, including active pSrc and pFAK. This Ambra1 binding-impaired mutant FAK protein caused increased steady-state levels of active pSrc and pFAK at focal adhesions, and enhanced adhesion, proliferation in 3D and chemotactic invasive migration, without affecting Src/FAK binding. Enhanced invasion was reversed by knockdown of the Ambra1-binding trafficking proteins Dynactin 1 and IFITM3, demonstrating that these Ambra1-scaffolded proteins are involved in the dynamic regulation of active pSrc and pFAK, presumably by trafficking them to and from focal adhesions. Indeed, we found that FAK-driven invasion in the SCC cells was hugely dependent on IFITM3, implying that its crucial role at the heart of trafficking protein networks linking Ambra1, FAK and active Src is vital to Src/FAK-mediated invasion. One caveat of interpreting the data with the FAK P875A/ P881A mutant protein is that this mutant is also impaired in binding p130Cas, although to a much

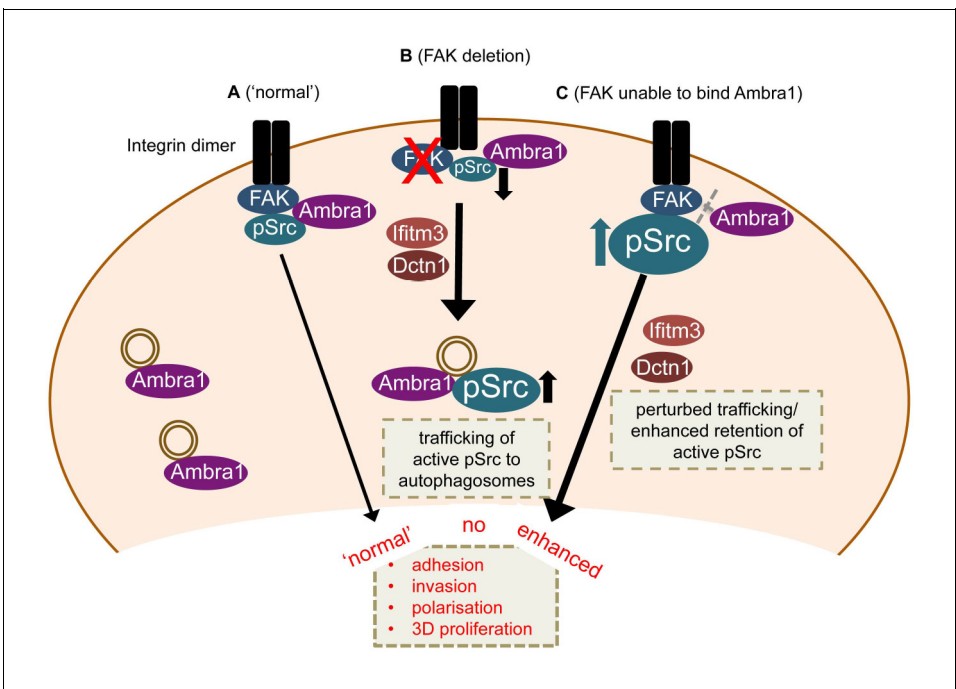

**Figure 7.** Model for Ambra1's role in pSrc trafficking and cancer cell phenotypes. In mouse SCC cells, Ambra1 is localised at autophagosomes and focal adhesions. (**A**) In FAK-WT expressing ('normal') SCC cells, Ambra1 binds to FAK and Src, regulating Src/FAK-mediated cancer processes like adhesion, invasion, polarisation and 3D proliferation. (**B**) In FAK -/- SCC cells, Ambra1 regulates the trafficking of active Src from focal adhesions to autophagosomes. Dynactin 1 (Dctn1) and IFITM3 are involved in this Ambra1-regulated trafficking process. (**C**) In cells expressing the FAK P875A/P881A mutant, Ambra1 binds to a lesser extent to FAK, but still to Src. This impaired FAK–Ambra1 interaction results in increased active Src levels at focal adhesions, resulting in enhanced adhesion, invasion, polarisation and 3D proliferation. Most likely this is due to perturbed trafficking of focal adhesion components, as knockdown of proteins involved in trafficking processes, like Dynactin 1 and IFITM3, rescues the phenotypes in FAK P875A/P881A expressing SCC cells.

lesser extent than Ambra1; importantly though, and in contrast to knockdown of Ambra1, knockdown of p130Cas did not significantly affect invasion (not shown).

In summary, we have identified a novel role for Ambra1 as part of an intracellular trafficking network of proteins that control the steady-state levels, and dynamics, of active Src and FAK at focal adhesions. Ambra1 scaffolds proteins such as Dynactin 1 and IFITM3 – and here, for the first time, we link these to the spatial control of active Src and FAK, regulating their steady-state levels at focal adhesions and the autophagic targeting of active Src when FAK is absent. As a result, Ambra1 and its interacting partners control cancer cell adhesion, polarisation, proliferation in 3D and chemotactic invasion. Deregulation of these trafficking components of focal adhesion complexes inhibits Src/FAK-dependent biological processes in cancer cells. Our work demonstrates the importance of tight dynamic control of trafficking of active Src, in particular, to and from focal adhesions.

## Materials and methods

### Antibodies, inhibitors and DNA constructs

Antibodies used were as follows: anti-Paxillin (RRID:AB_647289), anti-GM130 (RRID:AB_398141), anti-p130Cas (RRID:AB_397667) and anti-RACK1 (RRID:AB_397577) antibodies (BD Transduction Laboratories, New Jersey, USA), anti-IFITM3 (RRID:AB_2122095) (Abcam, Cambridge, UK), anti-CoxIV (RRID:AB_10694213), anti-FAK (RRID:AB_10694068), anti-pFAK Y397 (RRID:AB_2173659), anti-pPaxillin Y118 (RRID:AB_2174466), anti-Rab7 (RRID:AB_1904103), anti-pSrc Y416 (RRID:AB_331697), anti-Src (clone 36D10) (RRID:AB_10693939), anti-LC3B (RRID:AB_2137707) and anti-GAPDH (RRID:AB_10622025) (Cell Signaling Technologies, Danvers, MA, USA), and anti-Dctn1 (RRID:AB_10842517), anti-Ambra1 (RRID:AB_2636939) and anti-pSrc Y416 (RRID:AB_309898) antibodies and anti-FAK (clone 4.47)-conjugated agarose antibody (RRID:AB_310789) (Millipore, Billerica, MA, USA). LC3B antibody was from MBL (RRID:AB_1279144) (MBL International, Woburn, MA, USA). TRITC-phalloidin was purchased from Life Technologies (RRID:AB_2572408) (Paisley, UK). Anti-rabbit (RRID:AB_2099233) or mouse (RRID:AB_330924) peroxidase-conjugated secondary antibodies were purchased from Cell Signaling Technologies. Dasatinib was obtained from Bristol Myers Squibb (Princeton, NJ, USA) and PF562271 from Pfizer (Groton, CT, USA).

### Generation of FAK mutant constructs

FAK mutants were generated by site-directed mutagenesis using PFU Ultra Hotstart DNA polymerase (Stratagene, Amsterdam, The Netherlands) and the following primers (mutated base pairs are underlined): P875A (forward 5' – GATCATGCCGCTCCAGCAAAGAAGCCCCCT – 3', reverse 5' – GCGAGGGGGCTTCTTTGCTGGAGCGGCATG – 3') and P881A (forward 5' – CAAAGAAGCCCCCTCGCGCTGGAGCCCCCC – 3', reverse 5' – CAAGTGGGGGGCTCCAGCGCGAGGGGGCTT – 3'). After DpnI digestion for 1 hr at 37°C, chemically competent TOP10 bacteria were transformed.

### Cell culture and transfection

FAK-deficient SCC cell lines were generated, authenticated and characterised as described previously (*Serrels et al., 2012, 2010*). SCCs were maintained in Glasgow MEM containing 10% FCS, 2 mM L-glutamine, non-essential amino acids, sodium pyruvate and MEM vitamins at 37°C, 5% $CO_2$. SCC FAK-WT cells were maintained in 1 mg/ml hygromycin B. Ambra1 +/+ and -/- RasV12/E1A-transformed MEFs were a generous gift from Guillermo Velasco (*Cianfanelli et al., 2015*). MEFs were cultured in DMEM supplemented with 10% FCS and 2 mM L-glutamine at 37°C, 5% $CO_2$. Following thawing cells were used for no longer than three months. Original cells were pathogen tested using the ImpactIII test (Idexx Bioresearch, Westbrook, ME, USA) and were negative for all pathogens tested. All cell lines were routinely tested negative for mycoplasma contamination.

### siRNA

Ambra1 siRNA pool (cat. no. M-059556–01), individual Ambra1 siRNAs (cat. no. D-059556–01, 5' – GAAGAAUGCUGUACGAAUC – 3'; D-059556–04, 5' – CAACGUGCCCUCCUGCAAU – 3'), Dctn1 siRNA pool (cat. no. M-044821–01), IFITM3 siRNA pool (cat. no. M-056653–01) or scrambled siRNA (cat. no. D-001206-13-20) were purchased from Dharmacon (Loughborough, UK). FAK-WT or FAK -/- SCC cells were transiently transfected using HiPerFect (Qiagen, Manchester, UK) according to the

manufacturer's protocol, with a final concentration of 100 nM siRNA, respectively. Cells were analysed at 48–72 hr post transfection.

## Mapping of the Ambra1 binding site in FAK

The Ambra1 binding site in FAK was identified using peptide arrays as published previously (*Serrels et al., 2010*, *2007*). Briefly, overlapping 25-mer peptides of FAK were spotted onto nitrocellulose and incubated with recombinant Ambra1 (Origene, Herford, Germany). After extensive washes, the array was incubated with anti-Ambra1 antibody and then subjected to western blotting. For the identification of core amino acids, overlapping 25-mer peptides with one amino acid mutated at a time were used.

## Immunoblotting, immunoprecipitation and mass spectrometry

Cells were washed twice in ice-cold PBS and then lysed in RIPA buffer (50 mM Tris-HCl, pH 8.0, 150 mM NaCl, 1% Triton X-100, 0.1% SDS and 0.5% sodium deoxycholate) supplemented with PhosStop and Complete Ultra Protease Inhibitor tablets (Roche, Welwyn Garden City, UK). Lysates were cleared by centrifugation at 10000 rpm for 15 min and analyzed by western blotting. Protein concentration was calculated using a BCA protein assay kit (Thermo Scientific, Loughborough, UK). For immunoprecipitation, 1 mg cell lysates were incubated with 2 µg of unconjugated or 10 µl of agarose-conjugated antibodies at 4°C overnight with agitation. Unconjugated antibody samples were incubated with 10 µl of Protein A or Protein G agarose for 1 hr at 4°C. Beads were washed three times in lysis buffer and once in 0.6 M LiCl, resuspended in 20 µl 2x sample buffer (130 mM Tris-HCl, pH 6.8, 20% glycerol, 5% SDS, 8% $\beta$-mercaptoethanol, bromophenol blue) and heated for 5 min at 95°C. Samples were then subjected to SDS-PAGE analysis using the Bio-Rad TGX pre-cast gel system. Proteins were immunoblotted using the Bio-Rad Trans-blot Turbo transfer system, blocked in 5% BSA in TBST (TBS supplemented with 1% Tween-20), and incubated with primary antibody overnight at 4°C. Blots were washed three times in TBST, incubated with peroxidase-conjugated secondary antibody for 45 min at room temperature, washed as before, developed using Clarity Western ECL Substrate (Bio-Rad, Hemel Hempstead, UK) and imaged using a Bio-Rad ChemiDoc MP Imaging System (Bio-Rad, Hemel Hempstead, UK).

For mass spectrometry, SCC FAK-WT and -/- cells were lysed in RIPA buffer. For the immunoprecipitations, 2 mg cell lysates (samples in triplicates) were incubated with 2 µg of unconjugated antibodies (anti-Ambra1, anti-pSrc Y416 and rabbit-anti-IgG) or 10 µl of agarose-conjugated antibodies (FAK–agarose) at 4°C overnight with agitation. Unconjugated antibody samples were incubated with 20 µl of Protein A agarose for 1 hr at 4°C. Beads were washed twice in lysis buffer and twice in PBS. Protein complexes were subjected to on-bead proteolytic digestion, followed by desalting and liquid chromatography–tandem mass spectrometry as reported previously (*Turriziani et al., 2014*). The mass spectrometry proteomics data have been deposited to the ProteomeXchange Consortium via the PRIDE partner repository (*Vizcaíno et al., 2016*) with the dataset identifier PXD006002. Trafficking proteins were filtered from the total datasets and the interaction network analysis was performed using Cytoscape (RRID:SCR_003032).

## Immunofluorescence microscopy and image analysis

Cells were grown on glass coverslips for 24 hr and washed once in TBS prior to fixation (3.7% formaldehyde, 100 mM PIPES, pH 6.8, 10 mM EGTA, 1 mM MgCl$_2$ and 0.2% Triton X-100) for 10 min. Cells were subsequently washed twice in TBStx (TBS supplemented with 0.1% Triton X-100) and blocked in TBStx block (TBStx supplemented with 3% BSA). Primary antibodies were incubated in TBStx block overnight at 4°C, followed by three 5 min washes in TBStx, incubated with Alexa Fluor-labelled secondary antibodies diluted 1:200 in TBStx block (Life Technologies, Paisley, UK) and washed as before prior to being mounted in Vectashield mounting medium containing DAPI (Vector Labs, Peterborough, UK). Cells were imaged using a FV1000 confocal microscope (Olympus, Southend-on-Sea, UK).

For the Total Internal Reflection Fluorescence (TIRF) microscopy an inverted IX81 microscope (Olympus, Southend-on-Sea, UK) with a 150 × 1.45 NA UAPON TIRF objective using 491 and 561 excitation lines was used. Colocalisation was analyzed using the ImageJ plugin JaCoP (*Bolte and Cordelières, 2006*).

The intensity of pSrc or Paxillin at focal adhesions from three independent experiments (five cells per experiment; at least 10 focal adhesions per cells) was measured using ImageJ (RRID:SCR_003070).

## Focal adhesion isolation

Focal adhesion isolation was performed following the protocol described in ref. (*Kuo et al., 2012*). Briefly, cells were rinsed with PBS and incubated with TEA buffer (0.2 M triethanolamine, pH 8.0) for 5 min. To apply hydrodynamic force, the cells were rinsed with PBS for 10 s using a Waterpik dental flosser set at 2 (Waterpik, Reigate, UK). After another wash with PBS, the remaining attached focal adhesions were fixed for immunofluorescence analysis.

For western blot analysis, focal adhesions were crosslinked with 3 mM dimethyl 3,3'-dithiobispropionimidate (DTBP; Thermo Scientific, Loughborough, UK) for 30 min at room temperature and then quenched with 20 mM Tris-HCl, pH 8.0, for 5 min at room temperature. After another wash with 20 mM Tris-HCl, pH 8.0, cells were rinsed with ice-cold PBS, and 5 ml of focal adhesion extraction buffer (0.5% Triton X-100, 20 mM $NH_4OH$ in PBS) was added to the cells for 5 min. Cells were rinsed with PBS for 10 s using a Waterpik dental flosser set at 2, and the remaining attached focal adhesions were lysed in 2x sample buffer. Western blot bands were analysed by densitometry.

## Cell polarisation assays

Cell polarisation assessing the orientation of the Golgi apparatus in wounded cell monolayers was examined as described in (*Serrels et al., 2010*). Briefly, $3 \times 10^6$ cells were plated on fibronectin-coated coverslips in 12-well plates for 3 hr. The cell monolayer was wounded with a pipette tip, incubated in full SCC growth medium for 1.5 hr and then fixed and stained with anti-GM130 antibody.

## Invasion assays

Invasion was analysed as described in (*Serrels et al., 2010*). Briefly, growth factor-reduced Matrigel (BD Biosciences, Oxford, UK) was diluted 1:1 in cold PBS and allowed to set at 37°C in transwells. $2 \times 10^4$ cells were seeded onto the underside of the transwell. After 4 hr, the transwells were washed in PBS and placed into serum-free SCC growth medium. Full growth medium containing 10% FCS was added on top of the Matrigel. After 72 hr, cell invasion was assessed by staining with 5 µM calcein (Life Technologies, Paisley, UK) for 1 hr. Horizontal *z* sections through the Matrigel were acquired at 10 µm intervals with an Olympus FV1000 confocal microscope. The images were evaluated using ImageJ software.

## Adhesion assay

Cells were washed twice in PBS, resuspended in PBS and rotated for 45 min at 4°C. After that, 8000 cells/well were plated in serum-free medium on fibronectin-coated 96-well plates or on plastic at 37°C. Cells were fixed after 20 min, 1, 2 and 6 hr with 25% trichloracetic acid (TCA) and stored overnight at 4°C. Plates were washed five times with $H_2O$ and dried at 37°C. The fixed cells were stained with sulforhodamine B (SRB) (0.4% SRB in 1% acetic acid) for 30 min, washed five times with 1% acetic acid and dried at 37°C. The SRB was dissolved in 10 mM Tris, pH 10.5, for 2 hr at room temperature and the absorbance was read at 540 nm. Samples were normalised to the 6 hr time point and relative adhesion was calculated by setting the FAK-WT values to 1.

## 3D proliferation assay

Cells were resuspended in a 1.4% methylcellulose solution in growth medium, plated on a layer of 0.9% agarose and incubated at 37°C, 5% $CO_2$. After nine days, images were taken from 6–10 random fields (10x magnification) and colonies were counted.

## Statistical tests

For measuring the intensity of pSrc Y416 staining at focal adhesions (immunofluorescence), *n* = 5 (5 cells per experiment). For the invasion assay, the experiment was repeated with the following times: siAmbra1, *n* = 5; FAK P875A/P881A mutant, *n* = 8; siDctn1; *n* = 6 and siIFITM3, *n* = 6. All other experiments shown, *n* = 3. Quantification of internalised active Src was carried out by counting 100 cells and calculating the percentage. Error bars for the graphs showing the intensity of pSrc Y416

staining at focal adhesions and the invasion assays represent s.e.m. Error bars for all other experiments show s.d. Student's *t*-test was carried out to calculate the statistical significance.

## Acknowledgements

This work was funded by CR-UK Programme Grant (C157/A15703), European Research Council Advanced Investigator Grant (294440 Cancer Innovation) to M Frame and proteomics was funded by Science Foundation Ireland (14/IA/2395). We thank A von Kriegsheim for help with mass spectrometry. The Edinburgh Super-Resolution Imaging Consortium (ESRIC) provided the microscope platform for the TIRF microscopy and we thank Alison Dun and Rory Duncan for their assistance with this. Ambra1-deficient MEFs were provided by Guillermo Velasco (Complutense University, Madrid).

## Additional information

### Funding

| Funder | Grant reference number | Author |
| --- | --- | --- |
| Cancer Research UK | C157/A15703 | Margaret C Frame |
| European Research Council | Advanced Investigator Grant (294440 Cancer Innovation) | Margaret C Frame |
| Science Foundation Ireland | 14/IA/2395 | Amaya Garcia-Munoz |

The funders had no role in study design, data collection and interpretation, or the decision to submit the work for publication.

### Author contributions

CS, ES, Designed and performed the cell biology and biochemical experiments, and co-wrote the paper. All authors read and approved the final version of the paper; AB, Contributed to proteomics and network analysis, and edited the paper. All authors read and approved the final version of the paper; KP, Studied the biochemical complex between Ambra1 and FAK. All authors read and approved the final version of the paper; GSB, Mapped the Ambra1 binding site in FAK. All authors read and approved the final version of the paper; AG-M, CV, Provided reagents and advise. All authors read and approved the final version of the paper; FC, Contributed to proteomics and network analysis, and edited the paper. Mapped the Ambra1 binding site in FAK. Performed the initial screen and identified Ambra1 as a FAK interactor. All authors read and approved the final version of the paper; BS, Conceived the project, developed the proteomics, provided funding through competitively awarded grants and co-wrote the paper. All authors read and approved the final version of the paper; MCF, Processed the samples, analysed samples by LC-MS/MS, initial bioinformatics analysis (search and label-free quantification)

### Author ORCIDs

Christina Schoenherr, http://orcid.org/0000-0002-0983-6168
Adam Byron, http://orcid.org/0000-0002-5939-9883
Margaret C Frame, http://orcid.org/0000-0001-5882-1942

## Additional files

### Supplementary files

• Supplementary file 1. Ambra1 interacting proteins involved in trafficking. SCC FAK-WT and -/- cell lysates (in triplicates) were used for Ambra1-IP in order to determine specifically interacting proteins by quantitative label-free mass spectrometry. IgG served as a negative control. Mean mass spectrometry intensities of technical duplicate data acquisitions for each biological replicate are shown. Mean intensities for proteins not detected in either technical duplicate run were imputed with 1000. Peptide and protein false discovery rates were set to 1%. The mean intensities of Ambra1/IgG as well as Ambra1-IP SCC FAK-WT/SCC FAK -/- ratios were $\log_2$-transformed. The significance of

enrichment (Ambra1/IgG) was determined using two-tailed unequal variances $t$-test. Proteins were filtered according to their annotations with traffic-related gene ontology terms (transport, trafficking, vesicle, actin, tubulin and protein localisation).

• Supplementary file 2. FAK interacting proteins involved in trafficking. SCC FAK-WT and -/- cell lysates (in triplicates) were used for FAK-IP in order to determine specifically interacting proteins by quantitative label-free mass spectrometry. Mean mass spectrometry intensities of technical duplicate data acquisitions for each biological replicate are shown. Mean intensities for proteins not detected in either technical duplicate run were imputed with 1000. Peptide and protein false discovery rates were set to 1%. The mean intensities of FAK-IP SCC FAK-WT/SCC FAK -/- ratio were $\log_2$-transformed and the significance of enrichment (FAK-IP SCC FAK-WT/SCC FAK -/-) was determined using two-tailed unequal variances $t$-test. Proteins were filtered according to their annotations with traffic-related gene ontology terms (transport, trafficking, vesicle, actin, tubulin and protein localisation).

• Supplementary file 3. pSrc Y416 interacting proteins involved in trafficking. SCC FAK-WT and -/- cell lysates (in triplicates) were used for pSrc Y416-IP in order to determine specifically interacting proteins by quantitative label-free mass spectrometry. IgG served as a negative control. Mean mass spectrometry intensities of technical duplicate data acquisitions for each biological replicate are shown. Mean intensities for proteins not detected in either technical duplicate run were imputed with 1000. Peptide and protein false discovery rates were set to 1%. The mean intensities of pSrc/IgG as well as pSrc-IP SCC FAK-WT/SCC FAK -/- ratios were $\log_2$-transformed. The significance of enrichment (pSrc/IgG) was determined using two-tailed unequal variances $t$-test. Proteins were filtered according to their annotations with traffic-related gene ontology terms (transport, trafficking, vesicle, actin, tubulin and protein localisation).

### Major datasets

The following dataset was generated:

| Author(s) | Year | Dataset title | Dataset URL | Database, license, and accessibility information |
|---|---|---|---|---|
| Frame et al. | 2016 | Mass spectrometry proteomics data | http://proteomecentral.proteomexchange.org/cgi/GetDataset?ID=PXD006002 | Publicly available at ProteomeXchange (accession no: PXD006002) |

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
