## [Decision Letter]

Thank you for submitting your article "Ambra1 spatially regulates Src activity and Src/FAK-mediated cancer cell invasion via trafficking networks" for consideration by *eLife*. Your article has been favorably evaluated by Tony Hunter (Senior Editor) and three reviewers, one of whom is a member of our Board of Reviewing Editors. The reviewers have opted to remain anonymous.

The reviewers have discussed the reviews with one another and the Reviewing Editor has drafted this decision to help you prepare a revised submission.

Summary:

This study examines Ambra1 in a new role as a Src/FAK complex protein. The novel concepts being addressed are very interesting and potentially of relevance to the control of both adhesion signalling and autophagy.

Essential revisions:

The reviewers agreed that the study is potentially interesting and represents a novel role for Ambra1, but some aspects of the data require clarification and strengthening to support the conclusions being proposed. The reviewers agreed that the following points should be addressed in a revised manuscript:

1) The rationale for looking at Ambra1 is not explained, leaving the reader wondering why the authors landed upon Ambra1 as a modulator of FAK-dependent SRC activity? Did they perform a proteomic screen to look for FAK interacting proteins? This must be explained in more detail.

2) Co-IPs throughout the manuscript lack quantification and IgG controls are missing from Figure 1, Figure 3 – densitometry should be used to quantify given the central role of these experiments in the manuscript.

3) In Figure 1, Ambra1 is only detected at isolated focal adhesions and not in live cells – this discrepancy should be explained in the text.

4) Figure 1: A negative control (e.g. another autophagy protein) should be used to show the reported colocalization is not just accumulated cellular debris.

5) Co-localization of Ambra1 at focal adhesions in Figure 1 and Figure 2 should be quantified.

6) Cells in Figure 2 look morphologically distinct (larger nuclei etc.) after FAK depletion and indeed the most noticeable difference in Ambra1 is its increased nuclear staining. A relatively small proportion of Ambra1 is detected at autophagosomes in contrast to the large and increased amounts detected in the nucleus of FAK-/- cells. This should be discussed with respect to the authors proposed conclusions.

7) In several places, the authors use '% cells with internalised pSrc to make conclusions regarding active Src redistribution (Figure 2, Figure 4). The authors should analyze colocalization of pSrc with adhesions/autophagosomes and determine statistical significance of differences observed upon intervention, to support their comment directly on traffic from focal adhesions to autophagic structures.

8) Ambra1 appears to co-IP with both LC3B-I and LC3B-II, which is not consistent with it forming a complex at mature autophagosomes in which case, one would only expect to detect LC3B-II co-IPing with Ambra1. The LC3B input lanes in Figure 2 do not show LC3B-I and LC3B-II, and the poor quality of this blot makes the authors' conclusions dubious. Higher quality blots, quantification and re-wording of the potential interpretations should be included.

9) The Ambra-1 knockdown cells in Figure 2 should be co-stained for paxillin or some other focal adhesion complex component to robustly claim that Src is re-localizing to focal adhesions upon Ambra1 depletion (as is done in Figure 4).

10) Src levels in Ambra1 knockdown cells should be quantified as one would expect differences if Src is no longer trafficked to the autophagosome.

11) The significance of the FAK-/- cell findings would be greatly strengthened if Ambra1-dependent relocalization of Src to focal adhesions could be shown in detached cells.

12) The rationale for selecting dynactin1 and IFITM3 for further study out of the many proteins identified by mass spec as complexing with Ambra1, pSrc and FAK should be explained.

13) The potential for IFITM3 to localize to adhesions and the requirement for complex formation between pSrc and Ambra1 (within or outside adhesion complexes) should be tested? At present it is difficult to understand which proteins other than Ambra1 and FAK are directly interacting to mediate the specific effects.

14) The authors do not show co-localization of endogenous Ambra1 and FAK or Src in whole cells or in cells expressing the P875A/P881A mutant FAK that would justify their model. These data should be provided.

15) That Ambra1 is itself required for invasion, but a FAK mutant that does not bind Ambra1 promotes invasion is somewhat confusing. Both of these interventions result in increased pSrc at focal adhesions by disrupting the 'spatial rheostat'. The potential for Ambra1 compete with FAK for Src binding at adhesion and/or localize to adhesions in a FAK-independent manner should be discussed.

---

## [Author Response]

*Essential revisions:*

*The reviewers agreed that the study is potentially interesting and represents a novel role for Ambra1, but some aspects of the data require clarification and strengthening to support the conclusions being proposed. The reviewers agreed that the following points should be addressed in a revised manuscript:*

*1) The rationale for looking at Ambra1 is not explained, leaving the reader wondering why the authors landed upon Ambra1 as a modulator of FAK-dependent SRC activity? Did they perform a proteomic screen to look for FAK interacting proteins? This must be explained in more detail.*

We have explained this in more detail in the Results section (subsection “Ambra1 interacts with FAK in SCCs and localises at focal adhesions”, first paragraph). Essentially, we originally found Ambra1 in a phage display screen looking for novel FAK binding partners a few years ago, and this is now confirmed by reciprocal co‐IPs (Figure 1). As Ambra1 regulates autophagy, and we had already published that active Src was trafficked away from adhesions via autophagosomes in the absence of FAK (Sandilands et al., Nat Cell Biol, 2012), we set out to test the involvement of Ambra1 in the autophagic trafficking of active Src. This was the starting point for the present study.

*2) Co-IPs throughout the manuscript lack quantification and IgG controls are missing from Figure 1, Figure 3 – densitometry should be used to quantify given the central role of these experiments in the manuscript.*

We have added IgG controls for the FAK‐IPs in Figure 1 and Figure 3 and quantified all IPs by densitometry. Ratios are shown below the western blot panels or as graphs in the supplementary information.

*3) In Figure 1, Ambra1 is only detected at isolated focal adhesions and not in live cells – this discrepancy should be explained in the text.*

As Ambra1 shows dense diffuse intracellular staining, it was difficult to visualise any co‐localisation with FAK at focal adhesions in whole cells. Therefore, we isolated focal adhesions by hydrodynamic force. This has been re‐written in the manuscript (subsection “Ambra1 interacts with FAK in SCCs and localises at focal adhesions”, last paragraph). In addition though, we have now performed Total Internal Reflection Fluorescence (TIRF) microscopy (Figure 1—figure supplement 1; Figure 5—figure supplement 2), where we observed that Ambra1 was present at focal adhesions at the cell‐surface interface; specifically, there was partial overlap suggesting that, as with other autophagy proteins we have reported previously, there are pools of Ambra1 present at sub‐locations within focal adhesions.

*4) Figure 1: A negative control (e.g. another autophagy protein) should be used to show the reported colocalization is not just accumulated cellular debris.*

As many autophagosomal markers (e.g. Atg7, Atg12, LC3B) have been reported at focal adhesions (e.g. Sandilands et al., NCB 2012, Schoenherr et al., J Cell Sci 2014, Sharifi et al., Cell Rep 2016), we used a non‐autophagic, non‐adhesion cellular protein to demonstrate that the staining was not likely to be general cellular debris (Figure 1—figure supplement 1).

*5) Co-localization of Ambra1 at focal adhesions in Figure 1 and Figure 2 should be quantified.*

The co‐localisation of Ambra1 at focal adhesions has been quantified and COSTES r values have been added.

*6) Cells in Figure 2 look morphologically distinct (larger nuclei etc.) after FAK depletion and indeed the most noticeable difference in Ambra1 is its increased nuclear staining. A relatively small proportion of Ambra1 is detected at autophagosomes in contrast to the large and increased amounts detected in the nucleus of FAK-/- cells. This should be discussed with respect to the authors proposed conclusions.*

We do not see any consistent general differences in nuclear morphology or nuclear size. To be sure, we have now measured the size of the nuclei in FAK‐WT and ‐/‐ cells and could not detect any difference (Figure 8). However, we do believe that Ambra1 is probably also in the nucleus, since biochemical purification of nuclei, via fractionation and isolation through sucrose cushions, flowed by immunoblotting, suggests Ambra1 is in the nucleus irrespective of FAK status. We have added the latter evidence to Figure 2—figure supplement 1.

Author response image 1.Nuclear size comparisons in FAK-WT and FAK -/- cells.**DOI:**
http://dx.doi.org/10.7554/eLife.23172.025

The size of nuclei from 50 SCC FAK-WT and -/- cells was measured using ImageJ. Error bars, s.d.

*7) In several places, the authors use '% cells with internalised pSrc to make conclusions regarding active Src redistribution (Figure 2, Figure 4). The authors should analyze colocalization of pSrc with adhesions/autophagosomes and determine statistical significance of differences observed upon intervention, to support their comment directly on traffic from focal adhesions to autophagic structures.*

The co‐localisation of pSrc with Paxillin has been analysed, COSTES r values included and statistical tests performed (see Figure 2—figure supplement 1; Figure 4—figure supplement 1).

*8) Ambra1 appears to co-IP with both LC3B-I and LC3B-II, which is not consistent with it forming a complex at mature autophagosomes in which case, one would only expect to detect LC3B-II co-IPing with Ambra1. The LC3B input lanes in Figure 2 do not show LC3B-I and LC3B-II, and the poor quality of this blot makes the authors' conclusions dubious. Higher quality blots, quantification and re-wording of the potential interpretations should be included.*

We have included a higher quality blot and densitometric quantification of the LC3B‐IP (Figure 2). The reduced amount of LC3B‐II in SCC FAK ‐/‐ cells is consistent with previously published results (Sandilands et al., NCB 2012), although it is true to say that we don’t really know what this means for flux through the pathway of the Src‐selective autophagy. The antibody we used here immunoprecipitates both LC3B‐I and LC3B‐II. From these blots, we are not able to discern which isoform of LC3B is binding to Ambra1 or whether there is any difference between them. This is now stated in the text of the Results section (subsection “Ambra1 binds to Src and is required for trafficking of active Src to autophagosomes”, first paragraph).

*9) The Ambra-1 knockdown cells in Figure 2 should be co-stained for paxillin or some other focal adhesion complex component to robustly claim that Src is re-localizing to focal adhesions upon Ambra1 depletion (as is done in Figure 4).*

Paxillin staining has been included in the Figure 2.

*10) Src levels in Ambra1 knockdown cells should be quantified as one would expect differences if Src is no longer trafficked to the autophagosome.*

Ambra1 was knocked down with a pool or two individual siRNAs and pSrc was measured relative to protein loaded (using GAPDH as an invariant protein) by immunoblotting (Figure 2—figure supplement 1). Quantification is shown (Figure 2—figure supplement 1). As always, a significant reduction in the steady‐state level of cellular active pSrc correlates with autophagic removal from focal adhesions when FAK is absent. This is restored to equivalent FAK‐WT cell levels when Ambra1 is knocked down. Total Src levels are elevated in FAK ‐/‐ cells, and we have found this to be a transcriptional response to the loss of pSrc in FAK null cells via autophagy (not shown).

*11) The significance of the FAK-/- cell findings would be greatly strengthened if Ambra1-dependent relocalization of Src to focal adhesions could be shown in detached cells.*

Unfortunately, when the cells are detached, they do not have discernable focal adhesions and so it has not been possible to check for this. We have stated this in the text (subsection "Ambra1 binds to Src and is required for trafficking of active Src to autophagosomes”, last paragraph).

*12) The rationale for selecting dynactin1 and IFITM3 for further study out of the many proteins identified by mass spec as complexing with Ambra1, pSrc and FAK should be explained.*

We selected these two trafficking proteins to address the hypothesis that Ambra1 was influencing the localization of active pSrc via binding partners involved in trafficking. Initially we selected IFITM3 because it was present in complex with all three proteins as seen by interaction proteomics (Figure 3). We initially also selected Dynactin 1 as an example of an Ambra1‐interacting protein involved in trafficking, which, appeared to be enriched in FAK‐deficient cells in which pSrc was being trafficked via autophagy. In support of the hypothesis, knock down of both of these interacting partners of Ambra1 influenced the spatial localization of pSrc, allowing us to conclude that Ambra1’s role in pSrc cellular localisation is likely via trafficking complexes. We have now explained this more clearly in the text of the manuscript (subsection “Ambra1 is part of a network of trafficking proteins linked to active Src and FAK”).

*13) The potential for IFITM3 to localize to adhesions and the requirement for complex formation between pSrc and Ambra1 (within or outside adhesion complexes) should be tested? At present it is difficult to understand which proteins other than Ambra1 and FAK are directly interacting to mediate the specific effects.*

The IFITM3 antibody does not work well for immunofluorescence, which makes it difficult to show the localization of IFITM3 to focal adhesions. However, when we isolate focal adhesions by hydrodynamic force and analyse the lysates by immunoblotting, we detect IFITM3 in isolated focal adhesions (Figure 3—figure supplement 1). This implies that IFITM3 is likely localized at focal adhesions.

IFITM3 knockdown had no detectable impact on the Ambra1‐pSrc interaction (Figure 3—figure supplement 1), but knockdown of IFITM3 does cause a decrease in the interaction between pSrc and FAK (Figure 3—figure supplement 1), which likely reflects a requirement for optimal localisation of complex components.

*14) The authors do not show co-localization of endogenous Ambra1 and FAK or Src in whole cells or in cells expressing the P875A/P881A mutant FAK that would justify their model. These data should be provided.*

We have now added data to show that Ambra1, Src and FAK co‐localise at focal adhesions by using TIRF microscopy, to overcome the problem of much diffuse and generally vesicular Ambra1 staining (now shown in Figure 1—figure supplement 1; Figure 5—figure supplement 2).

*15) That Ambra1 is itself required for invasion, but a FAK mutant that does not bind Ambra1 promotes invasion is somewhat confusing. Both of these interventions result in increased pSrc at focal adhesions by disrupting the 'spatial rheostat'. The potential for Ambra1 compete with FAK for Src binding at adhesion and/or localize to adhesions in a FAK-independent manner should be discussed.*

Sorry, this was a bit confusing. To explain more clearly, Ambra1 is absolutely required for optimal invasion – since its depletion suppresses invasive migration. However, this is not linked to a greater amount of pSrc at focal adhesions in FAK‐WT cells as the reviewer suggests (see Figure 2). It is, however, linked to restoration of normal pSrc levels in FAK‐deficient cells, which are not invasive in any case. In the situation of the mutant FAK protein that cannot bind Ambra1 – the only case in which there is excessive active pSrc at focal adhesions – there is enhancedinvasion. Thus, elevated levels of active pSrc at focal adhesions are inextricably linked to increased invasive capacity.